# The follicle epithelium in the *Drosophila* ovary is maintained by a small number of stem cells

**Jocelyne Fadiga[1,2], Todd G Nystul[1,2]\***

[1]Department of Anatomy, University of California, San Francisco, San Francisco, United States; [2]Department of OB/GYN-RS, Center for Reproductive Sciences, University of California, San Francisco, San Francisco, United States

**Abstract** The follicle stem cells (FSCs) in the *Drosophila* ovary are an important experimental model for the study of epithelial stem cell biology. Although decades of research support the conclusion that there are two FSCs per ovariole, a recent study used a novel clonal marking system to conclude that there are 15–16 FSCs per ovariole. We performed clonal analysis using both this novel clonal marking system and standard clonal marking systems, and identified several problems that may have contributed to the overestimate of FSC number. In addition, we developed new methods for accurately measuring clone size, and found that FSC clones produce, on average, half of the follicle cells in each ovariole. Our findings provide strong independent support for the conclusion that there are typically two active FSCs per ovariole, though they are consistent with up to four FSCs per germarium.

## Introduction

The follicle epithelium of the *Drosophila* ovary has been a widely used and informative model for understanding epithelial tissue biology within the native, in vivo, environment (*Sahai-Hernandez et al., 2012*). First described over 60 years ago as a single layered epithelium that encapsulates developing germ cell cysts (*Demerec, 1950*; *King et al., 1956*), studies of this tissue have revealed insights into many aspects of epithelial biology, including diverse mechanisms that regulate the specification of cell fate in an epithelial stem cell lineage (*Assa-Kunik et al., 2007*; *Chang et al., 2013*; *González-Reyes and St Johnston, 1998*; *Johnston et al., 2016*; *Pocha and Montell, 2014*; *Song and Xie, 2003*), the establishment and maintenance of cell polarity (*Bilder et al., 2000*; *Castanieto et al., 2014*; *Kronen et al., 2014*; *Mirouse et al., 2007*; *St Johnston and Ahringer, 2010*), and the discovery of a novel mechanism for establishing planar polarity (*Cetera et al., 2014*; *Chen et al., 2016*).

A distinct advantage of the *Drosophila* ovary as an experimental model is that it has a highly consistent and well-described organization that facilitates the study of tissue biology with precise spatial and temporal resolution. Each *Drosophila* ovary is composed of long chains of developing follicles, called ovarioles (*Miller, 1950*), and oogenesis begins at the anterior tip of each ovariole in a structure called the germarium (*Koch and King, 1966*). The germarium has a stereotypical organization with four morphologically distinct regions, numbered from anterior to posterior as Regions 1, 2a, 2b, and 3 (*Figure 1—figure supplement 1A*). Germline stem cells (GSCs) reside at the anterior end of the germarium (*Carpenter, 1975*; *Koch and King, 1966*), in Region 1, and divide during adulthood to self-renew and produce daughter cells called cystoblasts. Cystoblasts undergo four rounds of mitosis with incomplete cytokinesis, as they move through Region 1 into Region 2a, which is defined by the presence of two 16 cell cysts that span the width of the germarium. Throughout Regions 1 and 2a, the germ cell cysts are covered by a population of somatic cells, referred to as inner

**\*For correspondence:**
todd.nystul@ucsf.edu

**Competing interests:** The authors declare that no competing interests exist.

germarial sheath (IGS) cells or escort cells. These cells provide a differentiation niche for the germ cells during these early stages of oogenesis (*Kirilly et al., 2011*), and may also help to propel the germ cells toward the posterior (*Morris and Spradling, 2011*). At the Region 2a/2b border, the cysts shed their IGS cell layer and move one at a time into Region 2b, where they become encapsulated by the follicle cell layer and take on a characteristic lens shape. Next, the cysts become more spherical in Region 3 (which is also referred to as Stage 1) and then bud off the germarium as a Stage 2 follicle. After budding, follicles rapidly grow and develop into a fully mature Stage 14 follicle that is ready for ovulation. This process, which takes approximately 8–9 days total under normal laboratory conditions (*King, 1970*), proceeds continuously during the first half of adult life, producing an organized tissue in which cells across the entire continuum of oogenesis are present simultaneously and arranged in order from the anterior to the posterior.

This thorough characterization of the ovarian structure and germ cell biology has facilitated the use of the ovary to study somatic cell biology as well. In a landmark study (*Margolis and Spradling, 1995*), the follicle epithelium was found to be maintained by a population of follicle stem cells (FSCs) in the germarium. This study was among the first to identify adult stem cells in vivo using site-specific DNA recombination to label clones of cells with a genetically heritable marker. In *Drosophila* studies, this is typically achieved by expressing flippase (Flp) to induce mitotic recombination between FRT sites on homologous chromosomes, whereas studies of mammalian tissues typically use cre-recombinase to induce intrachromosomal recombination at lox sites. This form of lineage tracing has become a gold standard for identifying adult stem cells (*Fox et al., 2008*; *Simons and Clevers, 2011*), and the methods used in the identification of FSCs helped to establish principles that have been applied to the study of many other Drosophila and mammalian adult stem cell lineages. Specifically, it is important to use a reliable clonal marking system in which the clonal marker is unambiguous, the method of clone induction does not significantly perturb tissue function or development, and the rate of background or 'leaky' clone induction is low. In addition, it is important to induce clones only sparsely, so that the majority of stem cells in the organ are not labeled with the clonal marker, to maximize the chance that each patch of labeled cells is one clone that originated from a single stem cell. With these quality control measures in place, the analysis of clone size and frequency at multiple time points can be used to infer the number of stem cells in the tissue as well as properties of the lineage such as the rate and number of transit amplifying divisions, the location of the stem cells, and the rate of stem cell turn over.

Using these principles, Margolis and Spradling induced sparse clones during adulthood and observed that single FSC clones contribute to an average of 50% of the follicle cell population on Stage 10 follicles, leading to the conclusion that each ovariole contains two actively dividing FSCs. In addition, they noticed that follicle cell clones remain coherent as a single patch of contiguous cells, even as the follicle grows and develops, and proposed that the FSCs reside at the Region 2a/2b border because persistent clones of contiguous cells at every time point always included one or more cells in this position. In contrast, a recent study concluded that there are 15–16 actively dividing FSCs per germarium that are organized into three rings, with one ring at the Region 2a/2b border, and two additional rings of cells in the adjacent positions immediately to the anterior, within Region 2a (*Reilein et al., 2017*). In addition, they proposed that, though all 16 FSCs are mitotically active, each FSC does not necessarily contribute to every follicle, so individual FSC clones can be discontinuous. The majority of the evidence in this study came from the use of a novel clonal marking system in which three clonal markers and two pairs of FRT sites are combined onto a single chromosome (*Figure 1—figure supplement 1B*). One homolog of Chromosome II has a constitutive GFP distal to FRT40A and a constitutive RFP distal to FRT42B, and the other homolog also has an FRT40A and an FRT42B with a constitutive Lac-Z that is distal to the FRT 40A. In this system, all cells initially express all three clonal markers, and heat shock-induced expression of Flp promotes recombination at one or both FRT sites, which has the potential to produce cells with 6 combinations of clonal markers (LacZ$^+$, GFP$^-$, RFP$^+$; LacZ$^+$, GFP$^-$, RFP$^-$; LacZ$^-$, GFP$^+$, RFP$^+$; LacZ$^-$, GFP$^+$, RFP$^-$; LacZ$^+$, GFP$^+$, RFP$^-$; LacZ$^+$, GFP$^+$, RFP$^+$) (*Figure 1—figure supplement 1B*).

Here, we evaluate the conclusions of the Reilein et al. study by testing the clonal system that was reported in this study and repeating key experiments from the Margolis and Spradling study using quantitative microscopy and two standard single color marking systems: a negatively marked clonal system and a MARCM system. We find that the multicolor system used in Reilein, et al. is highly unreliable, producing an unpredictable pattern of clones across multiple time points and a high rate

of 'leaky' clone induction in the absence of heat shock. In addition, we identify several logical flaws in the Reilein et al. study and performed new experiments to test the assumptions that underlie these flaws. Overall, our findings contradict the findings of Reilein et al. and instead support the original finding that the follicle epithelium is typically produced by two active FSCs residing at the FasIII border (*Margolis and Spradling, 1995*; *Nystul and Spradling, 2010*; *Nystul and Spradling, 2007*; *Spradling et al., 1997*). However, our quantitative approach revealed that the contribution from these two lineages is more variable than previously thought, leaving open the possibility that the number of active FSCs is not fixed but instead may fluctuate within a narrow range of one to four.

## Results

To test the LacZ, GFP, RFP (LGR) clonal marking system developed in Reilein, et al., we heat shocked the flies of the appropriate genotype for 1 hr at 37°C four times over 3 days, as described (*Reilein et al., 2017*), and assayed clone frequency at 5, 7, 9, 12, 14, 20, and 25 days post heat shock (dphs). Consistent with Reilein et al., we identified all six of the expected label combinations, and observed a progressive decrease in clonal diversity over the time course (*Figure 1A–D* and *Figure 1—figure supplement 2*). However, we observed less clonal diversity overall than reported in Reilein et al. For example, Reilein et al. reported that the majority of ovarioles had three uniquely labeled lineages at nine dphs, while the majority of ovarioles in our study had just two uniquely labeled lineages at this time point (*Figure 1E*). Notably, whereas the Reilein et al. study found a small number of ovarioles with five or six uniquely labeled lineages (8 and 4, respectively, out of 50 total at nine dphs), we never observed more than four uniquely marked clones within the same ovariole among the more than 300 ovarioles we imaged for this time course.

Importantly, we also noticed that, in the large majority (88.7%, n = 62) of ovarioles with more than two distinctly labeled lineages, only 1 or 2 of the lineages spanned the entire ovariole, while the other lineage(s) covered only part of one follicle (*Figure 1F*) or consisted of a small patch of cells in the germarium. This suggests that just one or two FSC lineages predominated in these cases. According to the criteria used in the Reilein et al. study (*Reilein et al., 2017*), the small clones in these ovarioles would also be considered FSC clones. However, there are several problems with this interpretation. First, it is generally inappropriate to use multiple heat shocks with a clonal marking system that has more than one pair of FRT sites because the later heat shocks may produce subclones within clones that were induced by the earlier heat shock treatments. Thus, at least some of the cases in which small and large clones coexist in the same ovariole may have been due to the production of subclones induced by the later heat shocks. Consistent with this, clone induction with a single 1 hr heat shock substantially reduced the number of distinctly labeled clones per ovariole at 9 and 14 dphs (*Figure 1G*). However, this may also be due to an overall reduction in the frequency of clone induction, and we still observed small and large clones coexisting in the same ovariole with this heat shock regimen in some cases.

Second, these small clones may have arisen independent of heat shock. To test this possibility, we compared the rate of clone induction with the LGR clonal marking system to a commonly used GFP negative clonal marking system with FRT19A (*Haelterman et al., 2014*; *Yamamoto et al., 2014*) in the absence of heat shock at 7 days after eclosion (*Figure 2A*). As expected, we found that the rate of clone induction was very low in the GFP negative system (<2%, n = 188). However, with the LGR clonal marking system, we found that, on average, 25% of ovarioles had FSC clones (with or without transient clones), and an additional 22% had transient clones without an FSC clone (n = 112) (*Figure 2A–B*). This contrasts with a frequency of 2% clone induction in the absence of heat shock in the Reilein et al. study. It is unclear why we observed different distributions of clone frequencies and rates of clone induction in the absence of heat shock, though differences in experimental conditions may have been a contributing factor. Nonetheless, our inability to reproduce their results suggests that a low rate of background clone induction, which is important for any clonal marking system, is not a robust feature of the LGR system.

Surprisingly, in flies with the LGR system, we identified clones that lacked expression of all three markers (*Figure 2C–D*) and others that expressed RFP but not LacZ or GFP (*Figure 2E*). This is unexpected because LacZ and GFP are both on the left arm of Chromosome II (2L), so all cells should express at least one of these two markers. We observed these types of clones in both heat-shocked

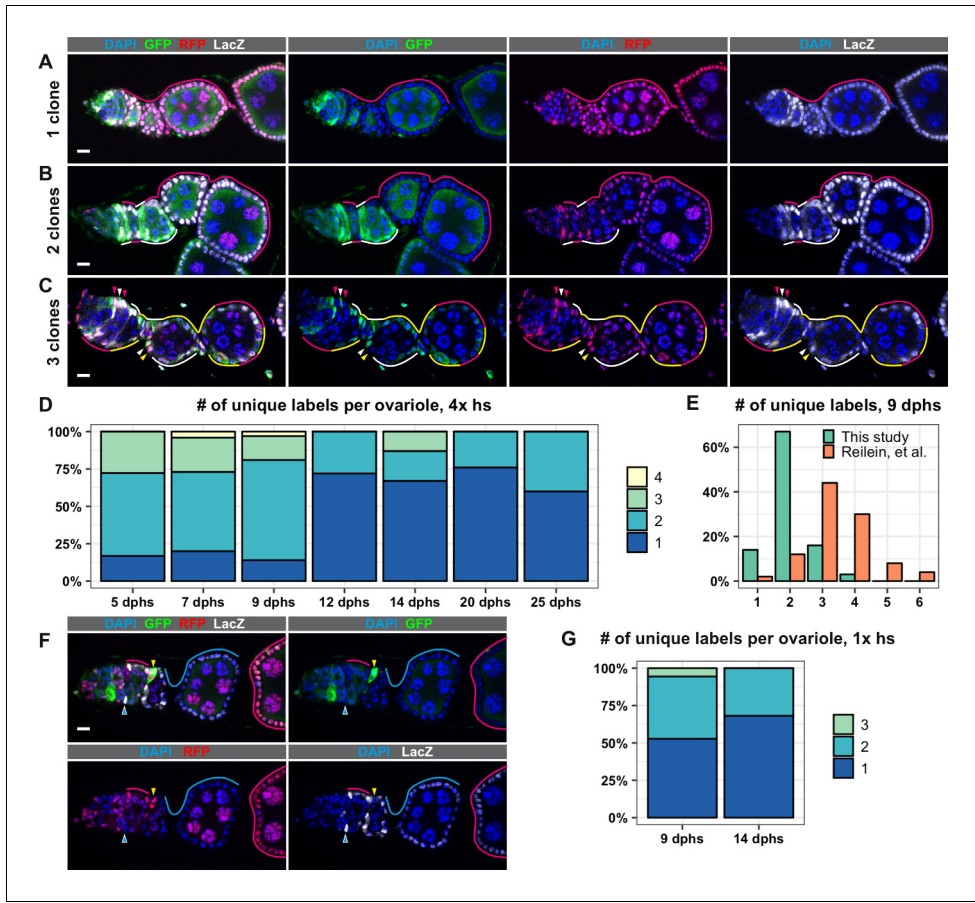

**Figure 1.** The LGR system labels clones with distinct marker combinations. (A–G) Analysis of LGR clones within the follicle epithelium, ranging from the Region 2a/2b border of the germarium to the first 2–3 follicles downstream from the germarium following four 1 hr heat shocks (A–F) or one 1 hr heat shock (G). (A–C) Examples of LRG clones at 7 days after the last heat shock (dphs) with 1 (A), 2 (B), or 3 (C) uniquely marked clones. Marker combinations present in these ovarioles are: LacZ⁺ GFP⁺ RFP⁺ (white lines and arrowheads), LacZ⁻ GFP⁺ RFP⁺ (yellow lines and arrowheads), LacZ⁺ GFP⁻ RFP⁺ (magenta lines and arrowheads). (D) Quantification of the number of uniquely labeled clones per ovariole after four 1 hr heat shocks at the indicated dphs, Total n = 242; five dphs n = 18, seven dphs n = 91, nine dphs n = 43, 12 dphs n = 36, 14 dphs n = 15, 20 dphs n = 29, 25 dphs n = 10. (E) Comparison of the number of the uniquely labeled clones per ovariole at nine dphs in our study (n = 43) and Reilein et al. (n = 50, taken from the Supplemental Note, Table a) plotted as a percent of total. (F) Example of an ovariole with two large clones that span multiple follicles (cyan triangle and cyan and magenta lines) and one small clone in the germarium (yellow triangles). (G) Quantification of the number of uniquely labeled clones per ovariole after one 1 hr heat shock at 9 and 14 dphs, Total n = 58; nine dphs n = 22, 14 dphs n = 36. Scale bars represent 10 μm.

The online version of this article includes the following source data and figure supplement(s) for figure 1:

**Source data 1.** Frequency of clones over a time course, LRG system, 4x heat shocks.
**Source data 2.** Comparison of this study with *Reilein et al. (2017)*.
**Source data 3.** Frequency of clones over a time course, LGR system, 1x heat shock.
**Figure supplement 1.** The *Drosophila* germarium and the LGR system.
**Figure supplement 2.** Individual channels of the four-color images.

and non-heat shocked cohorts. In all cases, other cells in the same ovariole clearly expressed all three markers, serving as an internal control for our ability to detect marker expression. In addition, we ruled out the possibility that the absence of LacZ and GFP signal was due to cell death because cells in the clone had normal nuclear morphology, and the clones often contained a dozen or more cells (*Figure 2D*), indicating that cells with these marker patterns were healthy enough to divide several times. We also ruled out the possibility that the absence of a LacZ and GFP signal was because

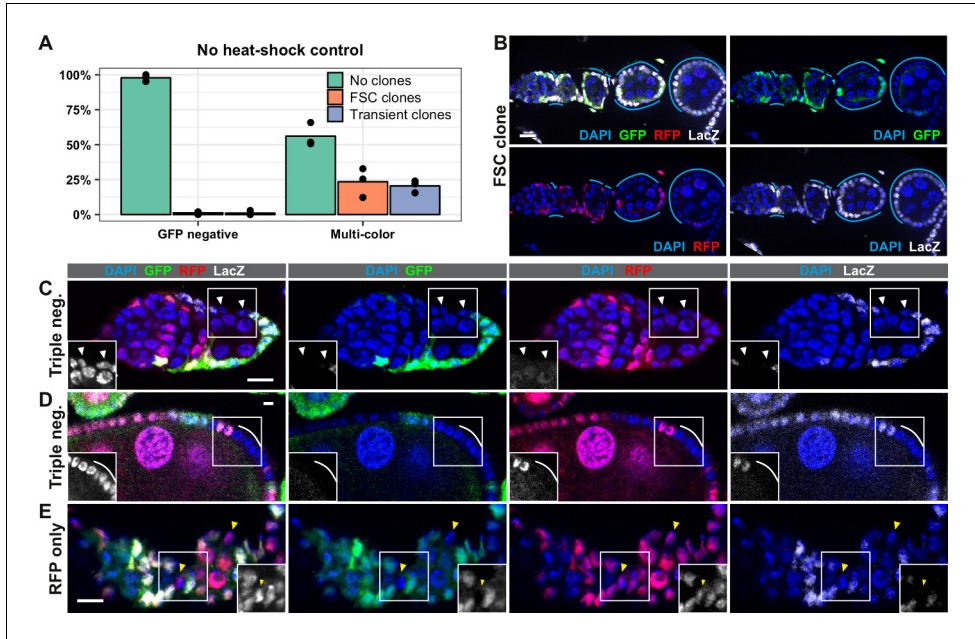

**Figure 2.** The LGR system is unreliable. (**A**) The frequency of ovarioles with either the negatively-marked FRT 19A system or the LGR system that have FSC clones (with or without transient clones), transient clones (without an FSC clone) or no clones at 7 days post eclosion in the absence of heat shock. Flies of both genotypes were maintained in vials from the same batch of food and kept in the same box at 25°C to minimize differences between cohorts, n = 153 for GFP negative system and n = 115 for LGR system. (**B**) Example of an ovariole with a large LacZ⁺ GFP⁻ RFP⁻ FSC clone from LGR flies that had not been heat shocked (cyan lines). (**C–E**) Examples of ovarioles with clones that have marker combinations that are not expected given the arrangement of the marker genes in the LGR system, including clones that are LacZ⁻ GFP⁻ RFP⁻ (white triangles and lines in C-D) and LacZ⁻ GFP⁻ RFP⁺ (yellow triangles in E). Scale bar represents 10 μm.

The online version of this article includes the following source data for figure 2:

**Source data 1.** No heat shock control, LGR system vs GFPneg system.

the cells were in mitosis, when nuclear markers are more difficult to detect, because the DNA did not appear condensed. In addition, we sometimes observed this clonal pattern in post-mitotic follicle cells (*Figure 2D*) and multiple adjacent cells, which are unlikely to all be in mitosis at the same time, shared the same marker pattern. Taken together, these findings reveal significant flaws with the LGR clonal system that severely reduce its reliability and reproducibility. This may account for a large part of the discrepancies between the Reilein et al. study and prior studies, and strongly suggests that the LGR system is not an appropriate tool for determining the number of FSCs per ovariole.

An assumption that was integral to the calculations of FSC number in the Reilein et al. study was that the clone induction protocol induced recombination at both FRT sites in every mitotic follicle cell. Specifically, they explain that recombination at both FRT sites followed by segregation of sister chromatids at mitosis would produce nine possible genotypes with equal frequency, and that these nine genotypes would produce six detectable phenotypes (marker combinations) at a 1:1:1:2:2:2 frequency. They then stated "We therefore assumed in our statistical modeling that the different colors of FSC clone were present in those same proportions (B:G:BG:BR:GR:LGR = 1:1:1:2:2:2)" (*Reilein et al., 2017*). However, we are not aware of any other example where recombination between FRT sites is reported to occur in every mitotic cell after heat shock. Cells must be in the S or G2 phase of the cell cycle for the recombination event to produce a detectable chromosomal rearrangement, and it is very unlikely that all cells will enter into this part of the cell cycle during the time that flippase is present after heat shock. Indeed, even in cases where the Flp is expressed constitutively, such as with *eyelss-Flp*, FRT recombination does not occur in all cells (*Menut et al., 2007*; *Moberg et al., 2005*). Using hs-Flp to generate negatively marked clones in the follicle epithelium, we typically find that only 25–50% of ovarioles have FSC clones (*Castanieto et al., 2014*;

*Johnston et al., 2016*; *Kim-Yip and Nystul, 2018*; *Kronen et al., 2014*). To investigate whether FRT recombination occurs in all cells in the LGR system, we quantified the frequency of each marker combination at multiple time points after either four 1 hr heat shocks or one 1 hr heat shock. We found that the most common marker combination at every time point was the unrecombined pheno-type, LacZ⁺, GFP⁺, RFP⁺, followed by LacZ⁺, GFP⁻, RFP⁺, while the remaining four marker combina-tions were present at much lower frequencies (*Figure 3A*). Unexpectedly, we found that clones marked by the absence of LacZ were less frequent than clones marked by the absence of GFP, indi-cating that cells lacking these two markers are not equally fit. These findings challenge the underly-ing assumptions of the analysis in the Reilein et al. study, and suggest that, even if the LGR system were reliable, the interpretation of the data leading to their conclusions about the number of FSCs per germarium are also flawed.

Therefore, to reexamine the original claims of the Margolis and Spradling study, we switched to the use of the FRT19A GFP negative clonal marking system that we found had a very low rate of background clone induction in the absence of heat shock (*Figure 2A*). Importantly, this system uses only one pair of FRT sites so there is no risk of generating subclones with multiple heat shocks. In addition, the generation of GFP negative clones with FRT19A is one of the most commonly used methods to label clones in studies of the FSC lineage (*Cook et al., 2017*; *Dai et al., 2017*;

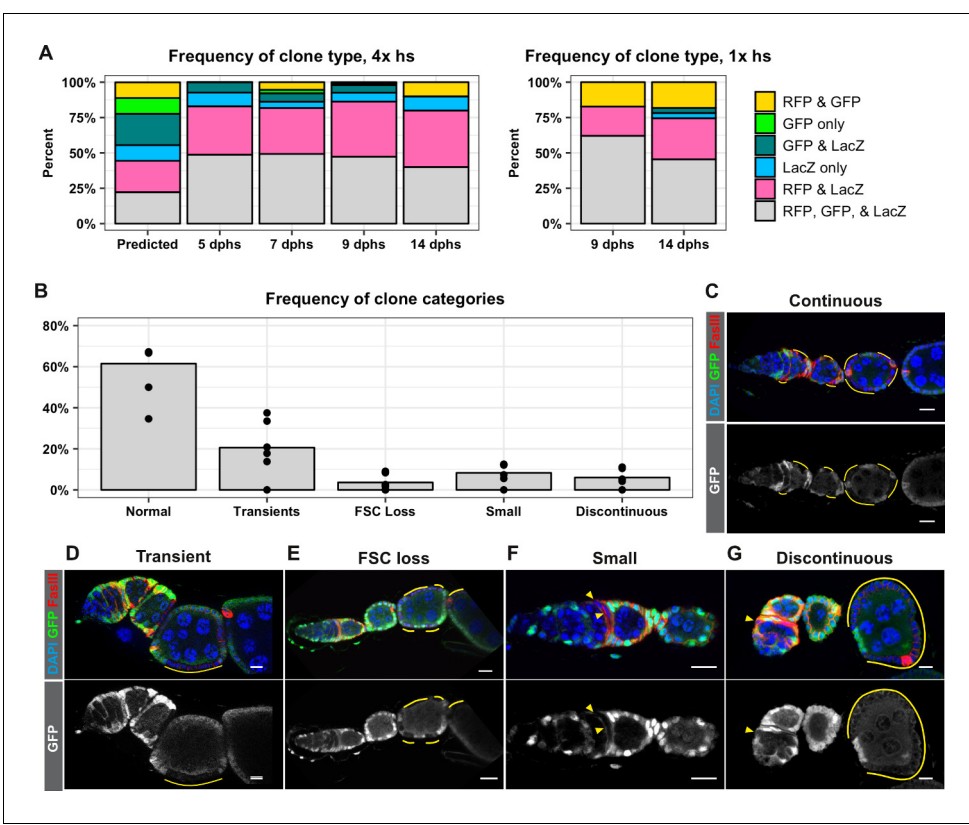

**Figure 3.** FRT recombination produces contiguous clones and does not occur in every cell. (**A**) A comparison of the frequency of each combination of clonal markers that would be predicted if recombination at both FRT sites occurred in all FSCs with the actual frequencies of clone types observed following either four 1 hr heat shocks or one 1 hr heat shock at the indicated dphs. For 4x hs, n = 242 and for 1x hs, n = 84. (**B–G**) A graph showing the frequencies of each type of clone observed with the negatively-marked FRT 19A system (B, n = 136) and examples of each clone type (**C–G**). GFP⁻ cells indicated by yellow lines and triangles. Scale bar represents 10 μm. The online version of this article includes the following source data for figure 3:

**Source data 1.** Frequency of each type of clone over a time course, LGR system, 4x heat shocks.
**Source data 2.** Frequency of each type of clone over a time course, LGR system, 1x heat shock.
**Source data 3.** Frequency of each type of clone pattern, GFPneg system.

*Nystul and Spradling, 2010*; *O'Reilly et al., 2008*). With this system, all cells initially express GFP, and FRT recombination produces clones that are clearly marked by the lack of GFP expression. One of the original observations in the Margolis and Spradling study is that an FSC clone forms a coherent, contiguous patch of cells that extends from the Region 2a/2b border out to the posterior edge of the clone boundary, contributing to each new follicle that is formed after clone induction. In contrast, Reilein et al. posited that each FSC does not normally contribute to every follicle, so discontinuous clone patterns with labeled cells at the Region 2a/2b border followed by intermittent, non-contiguous patches of labeled cells downstream should be expected. However, with the GFP negative clonal marking system, we found that the clone patterns in 94% of ovarioles with clones did not fit this expectation. Specifically, 61.5% of the ovarioles with clones had large FSC clones that extended from the Region 2a/2b border through the germarium and over one or more follicles (*Figure 3B–C*), 20.6% had transient clones that covered only a half of a follicle or less (*Figure 3B,D*), and 3.6% had large clones that spanned multiple follicles but did not extend back to the Region 2a/2b border (*Figure 3B,E*), which is the predicted pattern for a recent FSC replacement event (*Song and Xie, 2003*). The remaining 14.3% of ovarioles had one of two unexpected clone patterns: 8.3% had small clones within the germarium that typically covered only a small portion of one cyst (*Figure 3B,F*), and 6.0% had a discontinuous pattern in which a clone near the Region 2a/2b border coexisted with a larger, non-contiguous clone of cells further downstream in the ovariole (*Figure 3B, G*). These two clone categories occurred frequently enough that they seem unlikely to be due solely to background recombination events, though this remains a formal possibility. Alternatively, they may be evidence that FSC lineage dynamics are not as predictable as previously thought, and that rare events cause some cells to become quiescent or delay differentiation for a period of time. Therefore, although up to 6% of the ovarioles in this population may have had discontinuous clones, our results confirm that the large majority of FSC clones typically form coherent, contiguous units and contribute to every follicle within the boundaries of the clone.

Next, we sought to estimate stem cell number by measuring clone sizes at multiple time points. In the Margolis and Spradling study, this was achieved by measuring the size of clones on Stage 10 follicles, which provides a snapshot of the events that contributed to a single follicle in each ovariole. In that case, the authors reported that an average of 50% of the follicle cell population was labeled by each FSC clone. To assay FSC behavior during the production of multiple successive follicles, we used quantitative image analysis to count the number of GFP positive and GFP negative cells in mosaic ovarioles at two time points after a standard heat shock regimen of four 1 hr heat shock treatments over 2 days (*Figure 4A–B*). This heat shock regimen produced sparse clonal labeling, with only 33.9% and 35.0% ovarioles containing FSC clones at 7 and 14 dphs, respectively (*Figure 4C*). In mosaic ovarioles with GFP negative clones that extended from the Region 2a/2b border into at least one budded follicle, we observed an average clone size of 50.1 ± 14.1% at 7dphs and 51.3 ± 18.0% at 14dphs. We took two additional approaches to further test this result. First, we quantified FSC clone sizes at 7 days after a single 30 min heat shock. In these flies, the frequency of FSC clones was much lower (11.9%), yet the size of the FSC clones in mosaic ovarioles were not significantly different (48.2 ± 14.8%) from the average sizes of clones in mosaic ovarioles at 7 or 14 days after four 1 hr heat shock treatments (*Figure 4C–D*). Second, we repeated this test with another clonal marking system, MARCM (*Lee and Luo, 2001*), that uses a different mechanism to label clones. With this system, cells are initially GFP negative and mitotic recombination at FRT sites generates GFP positive clones. Again, we found that the average size of MARCM clones in mosaic ovarioles was not significantly different (44.7 ± 10.4%) from the average sizes of clones we observed with the GFP negative system (*Figure 4C–D*, and *Figure 4—figure supplement 1*). These measurements are remarkably similar to those reported in the Margolis and Spradling study, though we observed more variability in FSC clone size than was reported previously. Thus, our data provide strong support for the conclusion that the follicle epithelium is typically maintained by just two active FSCs. However, considering values that fall within two standard deviations of the means, and the fact that we excluded non-mosaic ovarioles from our analysis, our data are consistent with the possibility that FSC number fluctuates within a range of approximately one to four FSCs per ovariole.

Lastly, to determine where the FSCs are located, we stained ovarioles with GFP negative clones at 7dphs for GFP and FasIII. The Reilein et al. study reported that the anterior-most cells labeled with the LGR clonal marker system were located within one of three adjacent positions leading up to the FasIII border, and the authors interpreted this as evidence that FSCs are located within three

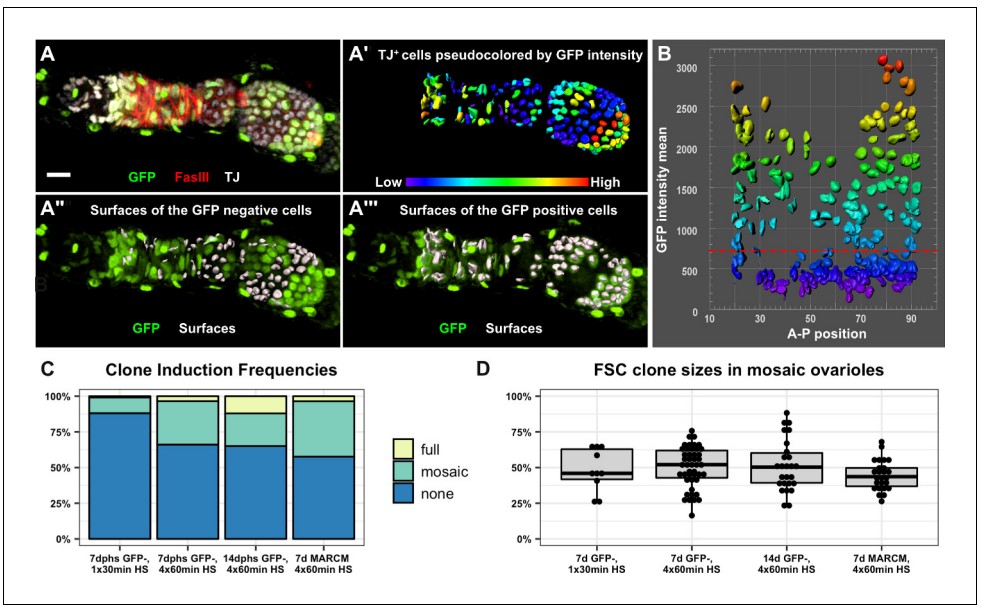

**Figure 4.** The follicle epithelium is typically maintained by two actively dividing FSCs. (**A**) 3D rendering of an ovariole stained for FasIII (red), GFP (green) and traffic jam (white). The traffic jam channel was used to generate surfaces around each somatic cell from the Region 2a/2b border to the posterior end of the clone (**A'**, surfaces are pseudocolored according to mean intensity in the GFP channel). GFP negative surfaces are shown with the GFP channel (green) in **A"** and GFP positive surfaces are shown with the GFP channel in **A'''**. (**B**) Surfaces were arrayed on a scatter plot according to the anterior/posterior position of each surface on the x-axis and the mean GFP intensity of each surface along the y-axis. The scatter plot was used to identify the threshold of GFP signal that differentiates between GFP negative and GFP positive cells (red dashed line). The 3D rendering (**A"** and **A'''**) was used to confirm the accuracy of the threshold and make adjustments to the threshold if necessary. (**C**) GFP negative FRT19A clones were induced with either one 30 min heat shock or four 1 hr heat shocks and analyzed at either 7 or 14 dphs. MARCM 19A clones were induced with four 1 hr heat shocks and analyzed at seven dphs. The percentage of ovarioles in which there are no FSC clones ('none'), an FSC clone that does not fully encompass each follicle ('mosaic'), and an FSC clone that fully encompases each follicle ('full') with the indicated experimental conditions. n = 243 for 30 min heat shock; n = 189 for 4 × 1 hr heat shock, 7dphs; and n = 181 for 4 × 1 hr heat shock, 14dphs; and n = 85 for MARCM19A. Clonal labeling was sparse, with no FSC clones in more than half of the ovarioles in all cases. (**D**) Quantification of clone size in mosaic ovarioles with continuous FSC clones extending from the Region 2a/2b border into the budded follicles indicates that, for all four experimental conditions, mean clone sizes are approximately 50% and are not significantly different from one another. For the GFP negative clones, n = 10 for the 30 min heat shock; n = 47 for 4 × 1 hr heat shock, 7dphs and n = 28 for 4 × 1 hr heat shock 14dphs. For MARCM clones, n = 25. p>0.5 for all pairwise comparisons. Scale bar represents 10 µm.

The online version of this article includes the following source data and figure supplement(s) for figure 4:

**Source data 1.** Frequency of clones in each experimental condition, GFPneg system and MARCM system.
**Source data 2.** Clone sizes in each experimental condition, GFPneg system and MARCM system.
**Figure supplement 1.** Quantification of MARCM clone sizes.

---

rings, two FasIII negative rings in Region 2a, and one ring at the boundary of FasIII expression. With the GFP negative clonal marking system, we found that the anterior-most GFP negative cell was FasIII positive and located at the boundary of FasIII expression in the large majority of ovarioles with FSC clones (85.7%, n = 63). In these germaria, all GFP negative somatic cells formed a single, contiguous clone and there were no GFP negative somatic cells anterior to the boundary of FasIII expression (*Figure 5A and D*), indicating that the FSC must be FasIII positive. This observation is consistent with other studies that have demonstrated that the anterior border of an FSC clone coincides with the boundary of FasIII expression (*Chang et al., 2013*; *Kirilly et al., 2005*; *Nystul and Spradling, 2010*; *Nystul and Spradling, 2007*; *Spradling et al., 1997*). In 7.9% of the ovarioles, the anterior-most GFP negative cell was contiguous with the rest of the clone but was located on the anterior side of the FasIII border. Thus, FasIII staining was detectable on the posterior surface of the cell (which is likely to be signal from the adjacent cell) but not the anterior surface (*Figure 5B and D*).

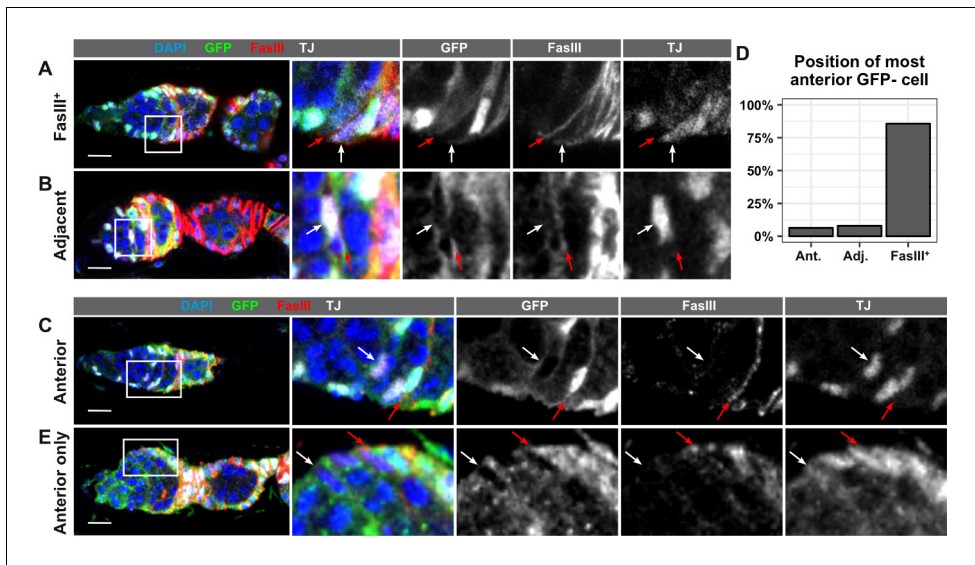

**Figure 5.** FSCs typically reside within the FasIII domain. (**A–C**) Ovarioles with GFP-negative FRT19A FSC clones stained for FasIII (red), GFP (green), and DAPI (blue) in which the anterior-most GFP-negative cell is FasIII+ and located at the boundary of FasIII expression (A, 'FasIII+'), FasIII− and located adjacent to the boundary of FasIII expression (B, 'Adjacent'), or FasIII− and anterior to the boundary of FasIII expression (C, 'Anterior'). (**D**) Quantification of the percentage of FSC clones in which the anterior-most GFP-negative cell is in each position, n = 61. (**E**) An ovariole with a GFP− cell in the 'anterior' position in a germarium without an FSC clone. White arrows indicate the anterior-most GFP− cell and red arrows indicate the boundary of FasIII expression. Scale bar represents 10 μm.

IGS cells in Region 2a are also mitotic during adulthood, and labeling systems that use mitotic recombination occasionally produce clones in Region 2a (*Kirilly et al., 2011*). Therefore, the anterior-most GFP negative cell in these germaria could be either an FSC with little or no FasIII expression or an IGS cell that became labeled through an independent recombination event from the one that produced the FSC clone. In the remaining 6.3% of ovarioles, the anterior-most GFP negative cell was FasIII negative and was not adjacent to either the FasIII border or the large clone of contiguous GFP negative follicle cells (*Figure 5C–D*). Notably, we found that cells in this position could be GFP negative even in germaria that did not have an FSC clone (*Figure 5E*), strongly suggesting that these cells are IGS cells, not FSCs. These observations confirm the conclusion that FSCs are usually, if not always, FasIII positive, and reside at the FasIII expression boundary.

## Discussion

An understanding about the number and position of FSCs in each germarium is an important foundation for the diverse range of studies that use the follicle epithelium as a model. Indeed, many studies have relied on the conclusions of the Margolis and Spradling study, and both the early descriptions of the ovariole and decades of subsequent work have reinforced these conclusions. For example, the prediction that there are just two active FSCs per germarium aligns well with the published rates of oogenesis and follicle cell number. Specifically, a careful measurement of cell number revealed that there are approximately 900 follicle cells per egg chamber at Stage 6, when the follicle cells exit mitosis (*Kolahi et al., 2009*), and measurements of the rate of oogenesis (*King, 1970*; *Lin and Spradling, 1993*; *Margolis and Spradling, 1995*) suggest that the transit time from the Region 2a/2b border to Stage 6 is approximately 100 hr. Two progenitor cells dividing continuously with a 9.6 hour cell cycle (*King and Vanoucek, 1960*; *Margolis and Spradling, 1995*) would take approximately 85 hr to produce 900 cells ($\log_2(900/2) \times 9.6 = 85$). Since the part of the lineage that differentiates into polar and stalk cells ceases division well before Stage 6, the actual time to produce 900 main body follicle cells would be somewhat longer. In contrast, 16 progenitor cells would

take only 56 hr (log$_2$(900/16) x 9.6 = 56) at this average rate of division. In addition, multiple studies have found that transient clones are capable of covering up to half of a follicle (*Nystul and Spradling, 2010*; *Skora and Spradling, 2010*; *Song and Xie, 2003*), consistent with the idea that the follicle cell population of each follicle is typically founded by two progenitors. Moreover, images of very large persistent clones from many different studies using a wide variety of single and multicolor labeling strategies and the analysis of the clonal data are consistent with a small number of actively dividing FSCs per germarium (*Dai et al., 2017*; *Kronen et al., 2014*; *Nystul and Spradling, 2007*; *O'Reilly et al., 2008*; *Song and Xie, 2002*). With respect to the position of the FSCs, the supposition that some cells in Region 2a are FSCs is inconsistent with many other studies that have demonstrated that somatic cells in this region have a markedly different shape, function, and gene expression pattern than cells in Region 2b (*Decotto and Spradling, 2005*; *Huang et al., 2014*; *Kirilly et al., 2011*; *Rust et al., 2019*; *Song and Xie, 2002*; *Spradling et al., 1997*; *Wang et al., 2015*; *Wang and Page-McCaw, 2018*; *Upadhyay et al., 2018*). Thus, the claim that there are 15–16 FSCs per germarium at both the Region 2a/2b border and within Region 2a is incompatible with a wide range of studies that provide independent orthogonal support for the conclusions of the Margolis and Spradling study.

Our study identifies several reasons for the discrepancies between the current paradigm and the Reilein et al study. First, their estimates of FSC number relied primarily on the use of the LGR clonal marking system, which we found to be unreliable because it had a high rate of background clone induction in our hands and produced clones with marker combinations such as LacZ$^-$, GFP$^-$, RFP$^+$ and LacZ$^-$, GFP$^-$, RFP$^-$ that are not predicted by the genotype. This raises the possibility that at least some of the clones generated by this system arise through a non-canonical event, such as recombination between an FRT40A and an FRT42D. This type of recombination would produce genome rearrangements that are not likely to be compatible with clone growth and, indeed, some clones of this type contained less than five cells. However, others, such as the clone shown in *Figure 2D*, were larger and the cells looked healthy. Thus, an alternative hypothesis is that these clones are not the product of an FRT recombination event but instead are due to epigenetic silencing of one or more marker transgenes. Consistent with this interpretation, high levels of fluorophore expression can reduce cellular fitness in some cases, so it may be that constitutive expression of three separate markers creates a selective pressure to silence one or more markers. Moreover, epigenetic silencing of transgenes has been described in follicle cells previously (*Skora and Spradling, 2010*). Although this study focused on transgenes with a UAS promoter rather than a constitutive promoter like those in the LGR system, it provides clear evidence that follicle cells actively silence transgene promoters and that the changes are heritable, producing a clone of cells with similar levels of transgene expression. Regardless of the cause, a high rate of clone induction in the absence of heat shock and the presence of clones with unexpected marker combinations are problematic for the use of clone size and frequency measurements to infer stem cell number. The Reilein et al. study also used a MARCM system as an independent test of their conclusions for this part of the study, but the number of clones analyzed was significantly lower and images of the MARCM clones were not provided so it is difficult to fully assess these data.

Second, we found that the assumptions and analysis of FSC clone number was flawed. The authors arrived at an estimate of 15–16 FSCs per germarium by quantifying the number of distinct lineages at 5, 7, 14, 21, and 30 dphs and then extrapolating back to 0 dphs using neutral drift modeling, but do not explain why the average number of uniquely labeled lineages would decline from more than four at five dphs to two at 15 dphs, and then remain at approximately two for the subsequent 15 days. Neutral competition models predict that mosaic tissues will drift toward monoclonality, not biclonality. Related to this, we found no evidence for the claim that individual FSC clones can be non-contiguous and contribute only sporadically to follicles that form after clone induction. On the contrary, we found that the large majority of FSC clones labeled with the GFP negative marking system in our study closely resembled the large contiguous clones described originally (*Margolis and Spradling, 1995*) and reported in many subsequent studies. Indeed, if FSC clones were typically discontinuous, it would be very difficult to distinguish between persistent FSC clones and transient clones because, even at late time points, it would appear as if 'transient' clones had not moved out of the tissue. Thus, much of the published work on persistent FSC clones would not have been possible (*Buszczak et al., 2009*; *Kirilly et al., 2005*; *O'Reilly et al., 2008*; *Song and Xie, 2003*; *Song and Xie, 2002*; *Su et al., 2018*). Therefore, rather than interpret the decrease in clonal

diversity at early time points in the Reilein et al. study as evidence for neutral drift among a large pool of FSCs, we favor the alternative hypothesis that the clonal diversity at early time points is due to the presence of transient clones that had not yet moved out of the tissue. Because the authors allowed for the possibility that FSC clones are not contiguous, these transient clones may have been misinterpreted as evidence for persistently labeled FSC lineages.

A parallel approach used in the Reilein et al. study to determine the number of FSCs per germarium was to induce LGR clones sparsely and measure the fraction of a follicle labeled by a rare FSC clone. This is a valid approach if one assumes that all stem cells in the tissue have an equal chance of becoming labeled. However, in Reilein et al., the authors argue that there are significantly different rates of proliferation among the 15–16 FSCs in each germarium. Since mitosis is required to label clones with the LGR system, more rapidly proliferating FSCs would be more likely to be labeled, though it would be difficult to predict how much the differences in proliferation rate would affect the chances of becoming labeled. The rapidly dividing FSCs would also presumably contribute a disproportionately high number of cells to the tissue, so the overrepresentation of clones from these FSCs in the dataset would skew the estimate of stem cell number by an unknown amount. Therefore, within the framework of the Reilein et al study, this would not be an appropriate approach to determine stem cell number.

Third, we challenged the assumption in the Reilein et al. study that FRT recombination at both sites occurs in 100% of mitotic follicle cells. This assumption does not accord with either our experience with the Flp/FRT system or the frequencies of marker combinations that we observed with the LGR system. Moreover, if both FRT sites always undergo recombination in every FSC, the same should occur in every mitotic follicle cell. Each ovariole has thousands of mitotic follicle cells, so at early time points such as 5 dphs, when cells produced in the germarium would not have moved out of the ovariole, every ovariole would have all six clonal marker phenotypes. Yet this clearly was not the case in either our study or in the Reilein et al. study. By assuming that the recombined phenotypes (marker combinations) should be more common than they actually are, the neutral drift analysis in Reilein et al. may have overestimated the amount of clonal diversity that was present prior to the first point of analysis (5 dphs), thus contributing to their overestimate of the number of FSCs.

Fourth, we used a standard, reliable clonal marking system and quantitative analysis to measure FSC clone size at two time points after clone induction. Our measurements were consistent with the original finding that each FSC typically produces approximately half of the follicle cells in each ovariole, though we found a wider range of clone sizes than expected. This finding, combined with the unexpected clone patterns that we observed (*Figure 3C and G–H*) suggest that FSC lineages are not rigid but instead exhibit some flexible, stochastic behaviors that allow for a variable number of FSCs or cause individual FSC lineages to contribute to the tissue unequally. An interesting possibility to consider in future studies is that unequal contributions between two stem cell lineages in the same germarium may precede an FSC loss and replacement event. Nonetheless, despite this variation in clone size and pattern, our findings reinforce the idea that two FSCs produce the large majority of the follicle epithelium in most cases.

Lastly, our analysis established that FSCs reside at the FasIII expression boundary. Somatic cells in Region 2a have a shape, function and gene expression pattern that is distinct from the follicle epithelium, so it would be surprising if the population of FSCs spanned this border, with some FSCs that have IGS cell characteristics and others that have follicle cell characteristics. Live imaging in the Reilein et al. study demonstrated that labeled cells near the Region 2a/2b border can move toward the anterior, but these cells could not be simultaneously labeled with cell type specific markers, so their identity is unclear. Further analysis will be required to determine whether there are any conditions in which cells convert from the IGS cell identity into a follicle cell identity or vice versa, but our data suggest that, at least under normal laboratory conditions, persistent FSC clones are typically contained entirely within the FasIII expression domain. Taken together, our study clarifies and reaffirms the current paradigm for the fundamental features of the *Drosophila* FSC lineage, and provides important context for a wide variety of studies that use the follicle epithelium as a model system.

# Materials and methods

**Key resources table**

| Reagent type (species) or resource | Designation | Source or reference | Identifiers | Additional information |
|---|---|---|---|---|
| Genetic reagent (*D. melanogaster*) | hsFlp; Ubi-GFP, FRT40a, FRT42D, Ubi-RFP | *Reilein et al., 2017* | | |
| Genetic reagent (*D. melanogaster*) | Tub-LacZ, FRT40a/CyO | *Reilein et al., 2017* | | |
| Genetic reagent (*D. melanogaster*) | y$^1$,w*, FRT 19A | *Yamamoto et al., 2014* | | |
| Genetic reagent (*D. melanogaster*) | w$^{122}$, hsFlp, Ubi-GFP, FRT 19A | *Yamamoto et al., 2014* | | |
| Genetic reagent (*D. melanogaster*) | hsFlp, tub-Gal80$^{ts}$, FRT19A; Act5C-Gal4, UAS-CD8::GFP/CyO | Built from BDSC stocks 5132 and 25374 | | |

## Fly stocks

Stocks were maintained on standard molasses food in an incubator at 25°C and adults were given fresh wet yeast daily. The following stocks were used:

1. hsFlp; Ubi-GFP, FRT40a, FRT42D, Ubi-RFP (*Reilein et al., 2017*)
2. Tub-LacZ, FRT40a/CyO (*Reilein et al., 2017*)
3. y$^1$,w*, FRT 19A (*Yamamoto et al., 2014*)
4. w$^{122}$, hsFlp, Ubi-GFp, FRT 19A (*Yamamoto et al., 2014*)
5. MKRS/TM6B, ry[CB] Tb[1] (BL #7304)
6. Canton S (BL#64349)
7. Oregon R-C (BL#5)
8. FRT 40A/CyO; D/TM3, ser
9. hsFlp; Ubi-GFP, FRT40a/CyO
10. hsFlp, tub-Gal80$^{ts}$, FRT19A; Act5C-Gal4, UAS-CD8::GFP/CyO

## Clone induction

Flies of the appropriate genotype were cultured and collected upon eclosion. Heat shocks were performed by transferring the flies to empty plastic vials and immersing them in a 37°C water bath for 1 hr, then returning the flies to vials containing food and wet yeast, and then placing them back in the incubator at 25°C. For generating LGR clones, this process was performed either four times over three days, as specified (*Reilein et al., 2017*) or once. For GFP negative clones, flies were heat shocked for 1 hr either four times over two days (once in the morning and once in the evening each day, with an approximately 8 hr interval between heat shocks) or just once for 30 min. For MARCM clones, flies were heat shocked for 1 hr four times over two days as specified for GFP negative clones, above. For the 'no heat-shock' controls, siblings of the flies that were exposed to heat shock were instead put directly in vials containing food and wet yeast in the incubator at 25°C. In all cases, after the heat shock regimen, flies were maintained in the incubator at 25°C and fed wet yeast daily until dissection. Flies were never kept outside of the incubator at 'room temperature' for extended periods of time because room temperature can vary from lab to lab and is inconsistent throughout the day and the year.

## Immunostaining

Ovaries were dissected in 1x phosphate buffered saline (PBS) or 1x Schneider's media, fixed in 1x PBS + 4% paraformaldehyde for 15 min, rinsed with 1x PBS + 0.2% Triton X-100 (PBST) and blocked for 1 hr with 1x PBST containing 0.5% BSA. Samples were incubated with primary antibodies diluted in blocking solution overnight at 4 deg C. Next, samples were rinsed with PBST and blocked for 1 hr before incubating with secondary antibodies for 4 hr at room temperature. Samples were rinsed twice with PBST and once with PBS before a final 30 min wash with PBS. Samples were mounted on glass slides in Fluoromount-G with DAPI (Fisher Scientific 00-4959-52).

The following primary antibodies were used: guinea pig anti GFP [1:1000] (Synaptic Systems 132005; RRID:AB_11042617), mouse anti-beta-Galactosidase [1:1000] (Promega Z3781; RRID:AB_430877), mouse anti-FasIII [1:100] (DSHB 7G10; RRID:AB_528238), rabbit anti-RFP [1:1000] (MBL

International PM005S; RRID:AB_591278), rat anti-RFP [1:1000] (ChromoTek 5F8; RRID:AB_2336064), rabbit anti-Castor [1:5000] (from Ward Odenwald) (*Kambadur et al., 1998*) guinea pig anti-traffic jam [1:5000] (from Dorothea Godt), and mouse anti-Groucho (DSHB AB_528272; RRID:AB_528272). The following secondary antibodies were purchased from Thermo Fisher Scientific and used at 1:1000: goat anti-guinea pig 488 (A-11073; RRID:AB_2534117), goat anti-rabbit 488 (A-11008; RRID: AB_143165), goat anti-rabbit 555 (A-21428; RRID:AB_2535849), goat anti-mouse 488 (A-11029; RRID:AB_2534088), goat anti-mouse 555 (A-21424; RRID:AB_141780), goat anti-rat 555 (A-21434; RRID:AB_2535855), goat anti-rabbit 633 (A-21070; RRID:AB_2535731), goat anti-mouse 633 (A-21050; RRID:AB_2535718), goat anti-guinea pig 633 (A-21105; RRID:AB_2535757), goat anti-mouse 647 (A-21236; RRID:AB_2535805), and goat anti-guinea pig (A-21450; RRID:AB_2735091).

### Image analysis

All fixed images were acquired using a Zeiss M2 Axioimager with Apotome unit, a Nikon C1 point-scanning confocal microscope, or a Nikon spinning disc microscope. For multicolor fluorescence images, each channel was acquired separately. Post acquisition processing such as image rotation, cropping, and brightness or contrast adjustment were performed using ImageJ and Photoshop. Acquisition settings and any brightness/contrast adjustments were kept constant across conditions within an experiment.

Clones were identified by acquiring stacks of optical sections through entire ovarioles and then stepping through the optical section to determine the marker status of each cell in the tissue, including markers in the far-red channel, which are not visible by eye under the microscope. This process was also used to identify the anterior-most cell in a clone and align the position of this cell with the FasIII boundary.

To automatically segment somatic cells, ovarioles were stained as described above for traffic jam or groucho to label somatic cell nuclei, and also for other proteins of interest (such as GFP or FasIII), and stacks of optical sections through entire ovarioles were acquired using a Nikon C1 confocal microscope. 3D confocal image stacks were processed in Imaris as follows. First, the image was rotated and voxels were remapped so the anterior/posterior axis of the ovariole was parallel to the x-axis of the image volume. Then, the traffic jam or groucho channel was processed for segmentation in Imaris. First the channel was duplicated and a gaussian filter was applied. Next, the sizes of several nuclei were measured, and the approximate nuclear diameter (usually ~2–2.25 µm) was used in the thresholding tool to subtract background. Lastly, the surface function was used to segment the processed channel. Within the surface function, a region of interest was defined, the local thresholding option was used, the split touching objects option was selected, and the quality threshold was adjusted to achieve, as close as possible, a single surface around each labeled nucleus. After the surface generation was completed, surfaces along the entire length of the ovariole were carefully inspected for errors. Single surfaces surrounding multiple nuclei were split into one surface per nucleus and surfaces that did not surround any nuclei were deleted. Surfaces were then used to quantify the signal intensities in other channels, such as the GFP channel.

To determine FSC clone sizes, ovarioles with clones that extended from the Region 2a/2b border to at least a Stage 2 follicle were selected for analysis. A scatterplot in the Vantage window of Imaris that arranged cells by A/P position along the x-axis and mean intensity of the GFP channel along the y-axis was examined to identify an approximate threshold between GFP$^-$ cells and GFP$^+$ cells. Typically, this was evident because GFP negative cells and GFP positive cells formed distinct clusters based on mean GFP intensity values. Cells close to the threshold on either side (GFP negative or GFP positive) were selected in the Vantage plot and checked by eye in the 3D view to confirm the GFP status of the cell. If errors were identified, the threshold was adjusted, and the process was repeated iteratively until an accurate threshold value was obtained. Then, the number of cells with a mean GFP intensity value below or above the threshold was used to calculate the total number of GFP negative and GFP positive cells in the region of interest.

### Statistics and graphs

Statistics and generation of graphs were performed using Excel (RRID:SCR_016137) or RStudio (RRID:SCR_000432). Raw data and source code for RStudio analysis are provided as supplemental files.

The equation used to estimate how long it would take a given number of FSCs to produce 900 cells is based on a standard equation for calculating doubling times. Specifically, in a cell lineage in which every cell divides into two cells at each generation, the number of cells, $P$, at the end of $n$ generations can be calculated with a simple exponential equation, $P = 2^n$. This equation can be generalized to account for the presence of multiple cell lineages in the same tissue that are all dividing at equal rates as follows: $P/l = 2^n$, where $l$ is the number of cell lineages in the tissue. To calculate how many generations ($n$) are required to reach a population size $P$ in a tissue with $l$ lineages, it is necessary to solve for $n$ by taking the $\log_2$ of each side of the equation: $\log_2(P/l) = n$. The rate of division, $r$, expressed as the amount of time per generation, ($r = t/n$), can be used to calculate how much time ($t$) it would take for a given number of FSCs to reach a given population size ($P$), as follows:

If $r = t/n$ then $n = t/r$

So, given $n = \log_2(P/l)$, then $t/r = \log_2(P/l)$, which simplifies to $t = \log_2(P/l) * r$

If $P = 900$ cells, $l = 2$ FSCs, and $r = 9.6$ hr/division, then $t = \log_2(900/2) * 9.6 = 85$ hr

If $P = 900$ cells, $l = 16$ FSCs, and $r = 9.6$ hr/division, then $t = \log_2(900/16) * 9.6 = 56$ hr

## Source code

Source code used to perform statistical analysis and generate graphs is provided as an R Notebook file. This code can be executed in R Studio by placing the file in the same directory as the Rdata file and running all chunks. R Studio will need to have the Tidyverse and Magrittr packages installed.

## Acknowledgements

We are grateful to Daniel Kalderon, Ward Odenwald, Dorothea Godt and the Bloomington Stock Center for stocks and reagents. We also thank Katja Rust, Sumitra Tatapudy and Nathaniel Meyer for helpful comments on the manuscript. This research was supported by a grant from the National Institute of Health: R01 GM097158.

## Additional information

### Funding

| Funder | Grant reference number | Author |
| --- | --- | --- |
| National Institutes of Health | GM097158 | Jocelyne Fadiga<br>Todd G Nystul |

The funders had no role in study design, data collection and interpretation, or the decision to submit the work for publication.

### Author contributions

Jocelyne Fadiga, Data curation, Formal analysis, Investigation, Visualization; Todd G Nystul, Conceptualization, Data curation, Software, Formal analysis, Supervision, Funding acquisition, Investigation, Visualization, Methodology, Project administration

### Author ORCIDs

Todd G Nystul (ID) https://orcid.org/0000-0002-6250-2394

### Decision letter and Author response

Decision letter https://doi.org/10.7554/eLife.49050.sa1
Author response https://doi.org/10.7554/eLife.49050.sa2

## Additional files

### Supplementary files

- Source data 1. RData file.
- Source code 1. R script file.

• Transparent reporting form

## Data availability

All data generated or analysed during this study are included in the manuscript and supporting files. Source data files have been provided for Figures 1-5.

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
