## [Decision Letter]

**Acceptance summary:**

This work revisited earlier studies regarding the behavior and characteristics of FSCs (follicle stem cells) in the *Drosophila* ovary, and identified a few potential flaws in the previous study, which concluded that there the number of FSCs was higher than previously thought. The current study presents data that are more in line with earlier studies that concluded that there are ~2 FSCs. The reviewers felt that the authors thoroughly examined the FSC clone behavior and provided potential explanations as to why previous and present studies differ in conclusion. The reviewers recognize that it may be sometimes difficult to reconcile differences between multiple studies, but the authors at least provided sufficient information such that this study itself can be thoroughly revisited/reproduced by anybody who wishes so. In summary, the reviewers agreed that this is an important study to characterize the clonal behavior of FSCs.

**Decision letter after peer review:**

Thank you for submitting your article "The follicle epithelium in the *Drosophila* ovary is maintained by a small number of stem cells" for consideration by *eLife*. Your article has been reviewed by three peer reviewers, one of whom is a member of our Board of Reviewing Editors, and the evaluation has been overseen by and Michael Eisen as the Senior Editor. The reviewers have opted to remain anonymous.

The reviewers have discussed the reviews with one another and the Reviewing Editor has drafted this decision to help you prepare a revised submission.

This manuscript by Fadiga and Nystul re-assessed the work by Reilein et al. from Kalderon laboratory regarding the characteristics and the behavior of follicle stem cells (FSCs) in the *Drosophila* ovary. There have been discrepancies in the field, mainly regarding the number and location of FSCs: former work suggested that there are ~2 FSCs/ovariole which locates at the boundary of region 2a/b, and Reilein et al. suggested that there are ~16 FSCs and some of them locate more anteriorly. By pointing out a few flaws in the previous study by Reilein et al., the present manuscript confirms the older work and conclude that there are only ~2 FSCs.

Overall, the reviewers found that the present work is generally convincing, but raised inter-related issues regarding whether or not this paper should be published in *eLife*. If the present study clearly points out the logical flaws and explains why the Reilein et al. study was incorrect, then the reviewers agreed that this study is worth publishing in *eLife*.

However, currently, this manuscript simply presents discrepancies between the present study and Reilein et al. paper, and if it is simply reporting discrepancies, it should be dealt as 'matters arising' 'communication' sort of publication at Nature Cell Biology, where Reilein et al. work was originally published.

To warrant the publication in *eLife* as a stand-alone work, the reviewers felt that the present work needs more experiments, so that it would go beyond simply reporting discrepancies and provide clearly higher level of rigor compared to Reilein work. To this end, the reviewers raised several important points to be addressed as following.

Major comments:

– The manuscript does not provide a clear view of the source of discrepancies between the present study and the Reilein et al. study, and it was very unclear whether the authors are claiming that Reilein et al.'s logic was flawed (e.g. insufficient control etc) or that Reilein et al.'s data and their data do not match. If the former, it can be strongly concluded that the present study is correct. But if the data do not match between independent experiments between two labs do not match, then, the readers need to be presented with additional evidence which study to be believed. For example, Figure 1E show that a simple discrepancy between two labs' data, without providing the reason why Reilein et al. was wrong. "We did not reproduce it" is not a strong enough argument to deny preceding work. In particular, it is well known that slight changes in experimental conditions influence the outcome of heatshock experiments (e.g. the temperature at which flies were raised prior to heatshock). Therefore, we cannot/should not scientifically judge who is right at this point. Similarly, although the present study concludes that multi-color lineage tracing system is unreliable using their own number of clone induction without heatshock (25%), Reilein et al. reports that such background clone induction is only 2%. Therefore, it is not that Reilein et al. was careless in doing their own experiments, and Reilein et al.'s interpretation itself was reasonable based on their own data. At least the authors have to precisely describe the nature of issues with the Reilein paper (flawed data interpretation vs. discrepancy with the author's own data). This confusion (unclear about flawed interpretation vs. discrepancy) is seem throughout the manuscript, and it has to be corrected such that the readers can understand exactly why the authors are stating Reilein et al., is not correct.

– If simply reporting discrepancies between authors' experiments vs. Reilein et al's experiments, it would be more suited for matters arising type publication at the journal of Reilein et al. (Nature Cell Biology). To make this paper beyond that level (i.e. worth of a stand-alone publication), the reviewers would like to see independent method to verify the authors' conclusion. In parallel, the reviewers noted that the heatshock treatments that span the duration of two days (for multicolor system), which were used by Reilein et al., as well as the present study, will produce many clones with considerably different birthtime. This will likely complicate the interpretation. Therefore, experiments that uses single heatshock is highly recommended. In this regard, this paper does not describe their heatshock scheme (how long, how many times) for any other heatshock experiments, thus it was hard for the reviewers to evaluate whether other experiments must give more reliable numbers or not.

– The reviewers discussed what would be the ideal 'independent' experiments that can provide independent testing of authors' vs. Reilein's conclusion. MARCM technique may be a choice (it still relies on mitotic recombination, but at least address the possibility of 'difficult to see' negative clonal marking). Additionally, FRT-stop-FRT type of constructs (which is already available: such has act-FRT-stop-FRT-lacZ or gal4) does not rely on mitotic recombination, thus it may be considered better as an independent method. Similarly, the use of Flybow system might work as well.

[Editors' note: further revisions were requested prior to acceptance, as described below.]

Thank you for submitting your article "The follicle epithelium in the  *Drosophila* ovary is maintained by a small number of stem cells" for consideration by *eLife*. Your article has been reviewed by two peer reviewers, including Yukiko M Yamashita as the Reviewing Editor and Reviewer #1, and the evaluation has been overseen by Michael Eisen as the Senior Editor.

The reviewers have discussed the reviews with one another and the Reviewing Editor has drafted this decision to help you prepare a revised submission.

The reviewers agreed that this is an important work trying to reconcile some discrepancies in the field regarding the number of follicle stem cells in *Drosophila* ovary. The authors did thorough analysis and identified a few possible flaws in the previous studies. Although this study might not completely settle the entire discrepancy, it at least provides a critical starting point for a better understanding.

As an essential revision, we would like you to present the following results more thoroughly, as these are critical information in assessing this work as well as comparing this work to others.

1) Multicolor labeling with a single heat shock

2) MARCM experiment

The authors may already have the results, but the results are not thoroughly described/discussed in the current manuscript.

Individual comments are provided as a guideline, but the above two points are the most critical ones.

Reviewer #1:

This is a revised manuscript from Nystul lab that tried to clarify the number of follicle stem cells (FSCs) in the *Drosophila* ovary. It has been postulated that there are ~2 FSCs/ovariole for a very long time, until a recent study proposed there are ~16 FSCs/ovariole. This manuscript re-evaluated the conclusion of the recent study to conclude that there is an average of ~2 FSCs (with variation range possibly 1-4 FSCs).

In the original manuscript, all the reviewers felt that it was unclear whether this manuscript should be published as a stand-alone paper, if it is only reporting a discrepancy from the previous paper. Reviewers commented that this paper will require clear explanation(s) as to why the previous paper might have been flawed to be acceptable in *eLife* as a stand-alone paper.

In this revised version, the authors generally made convincing case. However, I remain troubled by the use of multiple heatshock regimen by this paper as well as earlier Reilein paper. In my opinion, for multi-color system, multiple heatshock MUST NOT be applied, NEVER EVER. This is because later heatshock event will provide an opportunity to create 'subclones' within the original clone. For example, for the LGR (LacZ, Red, Green) scheme, the first heatshock may create LacZ, Green clone (losing Red). Then after this LacZ-Green clone proliferated a few times, second heatshock can hit to convert a subset of cells to become LacZ only (losing Green). If this happens, originally marked LacZ Green clone will be further subdivided to LacZ Green and LacZ only clone, as if there are two clones (instead of original one clone). For this reason, I have insisted for the longest time that multi color labelling should not be combined with multiple heatshocks (for single color, multiple heatshok is OK, because there is no chance to generate subclone within a clone'). I understand that the authors have already invested quite a bit of effort and time in this, but I need to insist NOT to use multiple heatshocks. The authors claims that this is 'to make the experimental condition consistent', but if such experimental condition is the very cause of logical flaw, it should not be repeated. Or at least, the results with single heatshock must be more thoroughly presented.

The presence of clones that lack both LacZ and GFP is indeed indicates a big flaw in the experimental scheme (I suspect that it may represent an event of recombination between FRT40 and FRT42).

The results of MRCM clone analysis must be more thoroughly showed, as this is provided as 'key experimental results' from an orthogonal testing.

Overall, I feel this is worth publishing, but the above comments should be addressed even if this goes to the second round of revision, because I think it is fair to provide more thorough information on this paper such that Reilein paper's authors can repeat and refute as necessary/appropriate, if they choose so. Although I think that it is generally not advisable or productive to allow multiple schools of scientists to keep refuting each other, sometimes it is inevitable, until the entire field settles. And at the least everyone should do their best to ensure everything (from reasoning to experimental condition) is clear to readers.

Reviewer #2:

I have seen the resubmission of the manuscript by Fadiga and Nystul readersand believe they have adequately addressed the reviewer's concerns. As I mentioned previously, I highly recommend publication of this paper, not only because it helps to clarify our understanding of FSC biology, but it also serves as an excellent primer for researchers from different backgrounds to learn how to properly carry out lineage studies.

1) In Figure 1D and 1E from what part of the ovary are the authors gathering data? The germarium? Region 2? Region 3? Stage 1? Stage 2?

---

## [Author Response]

This manuscript by Fadiga and Nystul re-assessed the work by Reilein et al. from Kalderon laboratory regarding the characteristics and the behavior of follicle stem cells (FSCs) in the *Drosophila* ovary. There have been discrepancies in the field, mainly regarding the number and location of FSCs: former work suggested that there are ~2 FSCs/ovariole which locates at the boundary of region 2a/b, and Reilein et al. suggested that there are ~16 FSCs and some of them locate more anteriorly. By pointing out a few flaws in the previous study by Reilein et al., the present manuscript confirms the older work and conclude that there are only ~2 FSCs.Overall, the reviewers found that the present work is generally convincing, but raised inter-related issues regarding whether or not this paper should be published in eLife. If the present study clearly points out the logical flaws and explains why the Reilein et al. study was incorrect, then the reviewers agreed that this study is worth publishing in eLife.

We thank the reviewers for their input and are pleased to hear that the reviewers agree that our work is generally convincing. Our study clearly identifies several logical flaws that may explain why the Reilein et al. study was incorrect. We have revised the text in several places to make this point clearer. The following sentences, some of which were also present in the original manuscript, are examples of places where we clearly indicate that our study identifies logical flaws in the Reilein et al. study:

1) In addition, we identify several logical flaws in the Reilein et al. study and performed new experiments to test the assumptions that underlie these flaws. Introduction paragraph five.

2) Taken together, these findings reveal significant flaws with the LGR clonal system that severely reduce its reliability and reproducibility. This may account for a large part of the discrepancies between the Reilein et al. study and prior studies, and strongly suggests that the LGR system is not an appropriate tool for determining the number of FSCs per ovariole. Results paragraph three.

3) They then stated “We therefore assumed in our statistical modeling that the different colors of FSC clone were present in those same proportions (B:G:BG:BR:GR:LGR = 1:1:1:2:2:2)”.

4) However, we are not aware of any other example where recombination between FRT sites is reported to occur in every mitotic cell after heat shock.

These findings challenge the underlying assumptions of the analysis in the Reilein et al. study, and suggest that, even if the LGR system were reliable, the interpretation of the data leading to the conclusion that there are 16 FSCs per germarium are also flawed. Results paragraph four.

5) One of the original observations in the Margolis and Spradling study is that an FSC clone forms a coherent, contiguous patch of cells that extends from the Region 2a/2b border out to the posterior edge of the clone boundary, contributing to each new follicle that is formed after clone induction. In contrast, Reilein et al. posited that each FSC does not normally contribute to every follicle, so discontinuous clone patterns with labeled cells at the Region 2a/2b border followed by intermittent, non-contiguous patches of labeled cells downstream should be expected. Using the GFP negative clonal marking system, we found that 85.7% of the ovarioles had typical clone patterns that had been described previously. Results paragraph five.

6) Thus, the claim that there are 16 FSCs per germarium at both the Region 2a/2b border and within Region 2a is incompatible with a wide range of studies that provide independent orthogonal support for the conclusions of the Margolis and Spradling study. Discussion paragraph one.

7) Our study identifies several reasons for the discrepancies between the current paradigm and the Reilein et al. study. First, their estimates of FSC number relied primarily on the use of the LGR clonal marking system, which we found has a high rate of background clone induction…Second, we found that the assumptions and analysis of FSC clone number was flawed…Third, we challenged the assumption in the Reilein et al. study that FRT recombination at both sites occurs in 100% of mitotic follicle cells. Discussions paragraphs one-three.

However, currently, this manuscript simply presents discrepancies between the present study and Reilein et al. paper, and if it is simply reporting discrepancies, it should be dealt as 'matters arising' 'communication' sort of publication at Nature Cell Biology, where Reilein et al. work was originally published.

Although we do report discrepancies between our study and the Reilein et al. study, we also used new approaches and experiments that went beyond simply repeating the experiments of Reilein et al.

These include the following:

1) The use of an independent and well-established clonal marking system, the GFP negative system on FRT19A, to confirm FSC clone size and dynamics.

2) The use of MARCM as a second, independent system, which we now include in response to reviewers’ comments. Our results provide further confirmation of our original conclusions about the number of FSCs per ovariole.

3) The use of 3D image segmentation methods to measure clone size as a percentage of total cell number in the tissue.

4) Quantification of images co-stained with FasIII and a clonal marker to precisely determine the position of the clone boundary relative to the FasIII expression boundary

To warrant the publication in eLife as a stand-alone work, the reviewers felt that the present work needs more experiments, so that it would go beyond simply reporting discrepancies and provide clearly higher level of rigor compared to Reilein work. To this end, the reviewers raised several important points to be addressed as following.Major comments:– The manuscript does not provide a clear view of the source of discrepancies between the present study and the Reilein et al. study, and it was very unclear whether the authors are claiming that Reilein et al.'s logic was flawed (e.g. insufficient control etc) or that Reilein et al.'s data and their data do not match. If the former, it can be strongly concluded that the present study is correct. But if the data do not match between independent experiments between two labs do not match, then, the readers need to be presented with additional evidence which study to be believed.

Although we cannot know precisely how or why the authors of the Reilein et al. study obtained each of the results they report, we have identified several instances in which the logic in the Reilein et al. paper is flawed and have pointed this out in our manuscript. These include:

1) The assumption that mitotic recombination produces clonal labeling in all mitotic cells. A significant flaw in the logic of the Reilein et al. study that was integral to their analysis was the assumption that their clone induction protocol induced recombination at both FRT sites in every mitotic follicle cell. Specifically, as we described in the manuscript, they explain that recombination at both FRT sites followed by segregation of sister chromatids at mitosis would produce 9 possible genotypes with equal frequency, and that these 9 genotypes would produce 6 detectable phenotypes (marker combinations) at a 1:1:1:2:2:2 frequency. They then stated “We therefore assumed in our statistical modeling that the different colors of FSC clone were present in those same proportions (B:G:BG:BR:GR:LGR = 1:1:1:2:2:2)”. There is no evidence that FRT recombination in a clonal marking system that relies on mitosis is 100% efficient in any fly tissue. Indeed, because cells must be in S phase or G2 for the recombination event to produce a detectable chromosomal rearrangement, it is extremely unlikely that all cells will be in the correct phase of the cell cycle at the time of the heat shock treatments to be susceptible to clonal labeling in this way.

2) The assumption that a quantification of the number of differentially labeled FSC clones per ovariole alone is sufficient to infer the number of FSCs per ovariole. This logic is stated in the Reilein et al. paper as follows: “The number of FSC colours observed (Figure 2J) and the inferred number of surviving FSC lineages (Figure 2C–K; see Supplementary Note) declined rapidly over time and, by extrapolation, indicates the initial presence of about 16 FSCs”. However, we found that, in 88.7% of ovarioles with more than two distinct lineages (n = 62), only 1 or 2 of the lineages spanned the entire ovariole while the other lineage(s) were very small, covering only part of one follicle or a small patch of cells in the germarium (Figure 1F). In Reilein et al., the authors did not comment on whether the clones they analyzed in this experiment were different sizes, so we do not know if this was also true in their hands, but our observations raise the possibility that they may have been. If clone sizes do differ in this way, it suggests that either the smaller clones are not FSC clones or that different FSCs within the same ovariole contribute substantially different numbers of cells to the tissue. Either of these possibilities, if true, would significantly undermine the logic used in Reilein et al. Specifically, if they interpreted non-FSC clones as FSC clones, this would lead to an overestimate of the number of FSCs in the tissue. Alternately, if some FSCs divide more often than others, then FSCs clones would be unequal sizes, which would contradict the assumption made elsewhere in the paper that clone size is proportional to stem cell number. Therefore, the failure to consider clone size in these experiments is a logical flaw of the Reilein et al. study that undermines their conclusions.

3) The assumption that FSC clones are commonly discontinuous. There is no evidence from the literature that this is the case and, in fact, several studies, including Margolis and Spradling, 1995, Nystul and Spradling, 2010, and Skora and Spradling, 2010 clearly demonstrate the contrary, that clones in the FSC lineage remain contiguous. In addition, as we described in the manuscript, if FSC clones were typically discontinuous, it would be very difficult to distinguish between persistent FSC clones and transient clones because, even at late time points, it would appear as if small “transient” clones had not washed out of the tissue. This would have made much of the published work on persistent FSC clones not possible. Lastly, we provide new evidence in this study that only 6% of ovarioles had clones that could be interpreted as discontinuous while 85.7% of the ovarioles had typical clone patterns that had been described previously. Interpreting patches of clonally marked cells that do not extend back to the Region 2a/2b border as FSC clones would lead to an inaccurate count of the number of FSC clones per ovariole. Therefore, the assumption that FSC clones are commonly discontinuous is a logical flaw of the Reilein et al. study that undermines their conclusions.

We have revised the text throughout the manuscript to make these points clearer.

For example, Figure 1E show that a simple discrepancy between two labs' data, without providing the reason why Reilein et al. was wrong. "We did not reproduce it" is not a strong enough argument to deny preceding work. In particular, it is well known that slight changes in experimental conditions influence the outcome of heatshock experiments (e.g. the temperature at which flies were raised prior to heatshock). Therefore, we cannot/should not scientifically judge who is right at this point. Similarly, although the present study concludes that multi-color lineage tracing system is unreliable using their own number of clone induction without heatshock (25%), Reilein et al. reports that such background clone induction is only 2%. Therefore, it is not that Reilein et al. was careless in doing their own experiments, and Reilein et al.'s interpretation itself was reasonable based on their own data.

We apologize for the confusion. By providing the comparison in Figure 1E, we did not intend to imply that our results with the LGR system should be considered to be more accurate than the results reported in Reilein et al. Instead, we provided this comparison to indicate that the LGR system, which they created and based most of their conclusions on, is apparently not reliable since we are unable to reproduce their results. In fact, we spent several months varying experimental conditions, such as culture temperature, heat shock conditions, and feeding, in an attempt to reproduce the results of the Reilein et al. study. However, in our hands, regardless of the experimental conditions we used, we consistently found that the numbers of clones labeled with different marker combinations were lower than those reported in Reilein et al. and the frequency of clones in the absence of heat shock was higher than the frequency reported in Reilein et al. In addition, when working with the LGR system, we were very surprised to discover that some cells could lack both LacZ and GFP expression (Figure 2C-E). Both of these markers are on the left arm of Chromosome 2 so, even after recombination, every cell should have at least one of these two markers. The emergence of LacZ-, GFP- clones is not likely to be due to variations in experimental conditions and instead suggests a fundamental flaw in the LGR system, thus further undermining its reliability.

Based on these observations, we concluded that the results with the LGR system from both our study and the Reilein et al. study could not be relied upon and thus a different clonal marking system that is more reliable and reproducible was necessary. We chose to work with the GFP negative marking system on FRT19A because it has been used by many different labs to study the ovary and we found the rate of background recombination to be very low (as is the case in the ovary for most GFP negative marking systems). We have now revised the text Results paragraph two and three to clarify our interpretation of our results with the LGR system. In response to the reviewers’ comments, we now indicate in this revised text that differences in experimental conditions may be one explanation for the differences in results between the two studies Results paragraph two. We hope this resolves the confusion about this issue.

At least the authors have to precisely describe the nature of issues with the Reilein paper (flawed data interpretation vs. discrepancy with the author's own data). This confusion (unclear about flawed interpretation vs. discrepancy) is seem throughout the manuscript, and it has to be corrected such that the readers can understand exactly why the authors are stating Reilein et al., is not correct. If simply reporting discrepancies between authors' experiments vs. Reilein et al's experiments, it would be more suited for matters arising type publication at the journal of Reilein et al. (Nature Cell Biology). To make this paper beyond that level (i.e. worth of a stand-alone publication), the reviewers would like to see independent method to verify the authors' conclusion. The reviewers discussed what would be the ideal 'independent' experiments that can provide independent testing of authors' vs. Reilein's conclusion. MARCM technique may be a choice (it still relies on mitotic recombination, but at least address the possibility of 'difficult to see' negative clonal marking). Additionally, FRT-stop-FRT type of constructs (which is already available: such has act-FRT-stop-FRT-lacZ or gal4) does not rely on mitotic recombination, thus it may be considered better as an independent method. Similarly, the use of Flybow system might work as well.

In response to this comment, we have now included an independent method to verify our conclusions. We considered each of the options recommended by the reviewers and decided to use MARCM because it is commonly used and well-tested; uses a single fluorophore and thus avoids potential issues with multicolor systems as we described for the LGR system used in Reilein et al.; and because, unlike Flpout systems, requires mitosis and thus is less prone to background recombination and labeling of non-mitotic cells that are not relevant for our analysis. We now include these data in an updated version of Figure 4C-D.

In parallel, the reviewers noted that the heatshock treatments that span the duration of two days (for multicolor system), which were used by Reilein et al., as well as the present study, will produce many clones with considerably different birthtime. This will likely complicate the interpretation. Therefore, experiments that uses single heatshock is highly recommended. In this regard, this paper does not describe their heatshock scheme (how long, how many times) for any other heatshock experiments, thus it was hard for the reviewers to evaluate whether other experiments must give more reliable numbers or not.

We completely agree with the concern that multiple heat shock treatments may complicate the analysis. We used four heat shock treatments to induce clones with the LGR system in order to stay consistent with the approach taken in Reilein et al. However, when we used the GFP negative system, we used two clonal induction regimens: In one case we used a single 30 minute heat shock treatment, and in a second case, we used four 1 hour heat shock treatments (Figure 4C-D). We found no significant difference in clone size using these two different protocols, which strongly supports the conclusion that the follicle epithelium is typically maintained by just two FSCs. We thank the reviewers for noticing that we did not fully explain this in the Materials and methods section. In response to this comment, we have now added details for each heat shock regimen and clonal marking system into the Materials and methods section.

[Editors' note: further revisions were requested prior to acceptance, as described below.]

The reviewers have discussed the reviews with one another and the Reviewing Editor has drafted this decision to help you prepare a revised submission.The reviewers agreed that this is an important work trying to reconcile some discrepancies in the field regarding the number of follicle stem cells in *Drosophila* ovary. The authors did thorough analysis and identified a few possible flaws in the previous studies. Although this study might not completely settle the entire discrepancy, it at least provides a critical starting point for a better understanding.As an essential revision, we would like you to present the following results more thoroughly, as these are critical information in assessing this work as well as comparing this work to others.1) Multicolor labeling with a single heat shock2) MARCM experimentThe authors may already have the results, but the results are not thoroughly described/discussed in the current manuscript.Individual comments are provided as a guideline, but the above two points are the most critical ones.

As requested, we now provide these data. The data for the use of multicolor labeling with a single heat shock are in Figures 1G and 3A. The new data from the MARCM experiment are in Figure 4—figure supplement 1.

Reviewer #1:

[…] In this revised version, the authors generally made convincing case. However, I remain troubled by the use of multiple heatshock regimen by this paper as well as earlier Reilein paper. In my opinion, for multi-color system, multiple heatshock MUST NOT be applied, NEVER EVER. This is because later heatshock event will provide an opportunity to create 'subclones' within the original clone. For example, for the LGR (LacZ, Red, Green) scheme, the first heatshock may create LacZ, Green clone (losing Red). Then after this LacZ-Green clone proliferated a few times, second heatshock can hit to convert a subset of cells to become LacZ only (losing Green). If this happens, originally marked LacZ Green clone will be further subdivided to LacZ Green and LacZ only clone, as if there are two clones (instead of original one clone). For this reason, I have insisted for the longest time that multi color labelling should not be combined with multiple heatshocks (for single color, multiple heatshok is OK, because there is no chance to generate subclone within a clone'). I understand that the authors have already invested quite a bit of effort and time in this, but I need to insist NOT to use multiple heatshocks. The authors claims that this is 'to make the experimental condition consistent', but if such experimental condition is the very cause of logical flaw, it should not be repeated. Or at least, the results with single heatshock must be more thoroughly presented.

We completely agree with the reviewer’s point that using multiple heat shocks with a clonal system that has more than one pair of FRT sites is problematic. In addition to adding new data for the use of multicolor labeling with a single heat shock in Figures 1G and 3A, we now also make this point in the text in the second paragraph of the Results section. Specifically, we state “…it is generally inappropriate to use multiple heat shocks with a clonal marking system that has more than one pair of FRT sites because the later heat shocks may produce subclones within clones that were induced by the earlier heat shock treatments. Thus, at least some of the cases in which small and large clones coexist in the same ovariole may have been due to the production of subclones induced by the later heat shocks.”

The presence of clones that lack both LacZ and GFP is indeed indicates a big flaw in the experimental scheme (I suspect that it may represent an event of recombination between FRT40 and FRT42).

We agree.

The results of MRCM clone analysis must be more thoroughly showed, as this is provided as 'key experimental results' from an orthogonal testing.

We now provide representative images of a 3D rendering of an ovariole with a MARCM clone, including the surfaces generated with Imaris and the associated scatter plot as Figure 4—figure supplement 1.

Reviewer #2:

I have seen the resubmission of the manuscript by Fadiga and Nystul readersand believe they have adequately addressed the reviewer's concerns. As I mentioned previously, I highly recommend publication of this paper, not only because it helps to clarify our understanding of FSC biology, but it also serves as an excellent primer for researchers from different backgrounds to learn how to properly carry out lineage studies.1) In Figure 1D and 1E from what part of the ovary are the authors gathering data? The germarium? Region 2? Region 3? Stage 1? Stage 2?

These data are derived from an analysis of the germarium and the first 2-3 follicles downstream from the germarium. We have added this detail into the figure legend for Figure 1.